# Microalgal Peloids for Cosmetic and Wellness Uses

**DOI:** 10.3390/md19120666

**Published:** 2021-11-26

**Authors:** M. Lourdes Mourelle, Carmen P. Gómez, José L. Legido

**Affiliations:** FA2 Research Group, Applied Physics Department, University of Vigo, 36310 Vigo, Spain; carmengomez@uvigo.es (C.P.G.); xllegido@uvigo.es (J.L.L.)

**Keywords:** peloids, microalgae, cyanobacteria, cosmetics, dermocosmetics, mineral water, seawater

## Abstract

Peloids have been used for therapeutic purposes since time immemorial, mainly in the treatment of locomotor system pathologies and dermatology. Their effects are attributed to their components, i.e., to the properties and action of mineral waters, clays, and their biological fraction, which may be made up of microalgae, cyanobacteria, and other organisms present in water and clays. There are many studies on the therapeutic use of peloids made with microalgae/cyanobacteria, but very little research has been done on dermocosmetic applications. Such research demonstrates their potential as soothing, regenerating, antioxidant, anti-inflammatory, and antimicrobial agents. In this work, a method for the manufacture of a dermocosmetic peloid is presented based on the experience of the authors and existing publications, with indications for its characterization and study of its efficacy.

## 1. Introduction

Peloids are therapeutic agents used in spas and thalassotherapy centers since time immemorial, mainly for treatment of osteo-articular and dermatological disorders, sports injuries, and generally in rehabilitation programs. Their use in cosmetics also dates back a long time, especially the ones made from clays, which are used in wellness programs and thermal spa centres nowadays [1]. 

Peloids are comprised of a solid fraction that includes various sediments, clays and peat, and a liquid fraction that can be either mineral-medicinal water (mineral water), seawater, or salt/brackish lake water. A biological fraction, consisting of microbiota present in mineral-medicinal water, clays, peat or sediments, and the microorganisms that thrive in the mixture during the peloid maturation processes, can also be present [2]. It is precisely during this maturation process (prolonged contact between solid substrate and liquid) that the different biological action compounds, partly responsible for the therapeutic actions, are formed [3].

Peloids either form “in situ” through contact between the mineral-medicinal water and the sediments surrounding it or are prepared artificially by mixing the above components [4]. When preparing peloids artificially, the biological fraction (microalgae and cyanobacteria) is usually from the natural mineral-medicinal water, while in the case of marine silt peloids, it is from cultivated microalgae; maturation times vary from 1–18 months but usually do not exceed 3 months [5]. According to Gomes et al. (2013), peloids can be classified regarding their origin, composition, and applications into “natural peloid” or “peloid *sensu strictu*”; “inorganic,” “organic,” or “mixed peloids”; and “medical” or “cosmetic” peloids (Figure 1) [4]. During the 3rd Symposium on Thermal Mud, held in 2004 in Dax, it was agreed to distinguish between the two main types of peloids: (i) muds or clays that are just mixed with mineral water with no maturation process—the extemporaneous or prepared *ad hoc* peloids—and (ii) muds or clays mixed with mineral water, including naturally or artificially matured peloids. Figure 2, Figure 3, Figure 4 and Figure 5 show two different types of natural-maturation and artificial-maturation peloids [4].

In order to evaluate peloid suitability for therapeutic and cosmetic purposes, the thermal properties of the mixtures like density, specific heat, thermal conductivity, and retention capacity are studied, as well as other properties related to applicability such as viscosity and pH [6]. 

Applicability, spreadability, user, and efficacy tests should be performed when used in dermocosmetics and/or wellness programs in thermal spas and thalassotherapy centres. Marketing of cosmetic peloids must comply with national legislation, which usually includes safety reports, user and efficacy tests, etc.

Thermal spas and wellness centers seldom use microalgae peloids in dermocosmetics, which is why this study reviewed such peloids and proposed a method for their manufacture. Therefore, the intention was to encourage spas to manufacture their own products for use in cosmetic and wellness applications through the required research and experience.

## 2. Peloids for Dermocosmetics and Wellness

For this review, SciFinder, Pubmed, Web of Science, and Scopus databases were reviewed up to September 2021. Search terms included “pelotherapy,” “mud therapy,” “peloids and skin,” “thermal mud,” “microalgae and thermal water,” “cyanobacteria and thermal water,” mud and cosmetics,” “mud and dermocosmetics,” “mineral water and skin,” and “seawater and skin”.

Although frequently used empirically, peloids have important cosmetic actions, which are linked to the improvement of skin hydration, the removal of flaking cells, and the prevention of aging [7,8,9]. Traditionally, the types of peloids most used in cosmetics are volcanic, sulphurous, and chlorinated bromo-iodics, but this also includes peat due to its content of fulvic and ulmic acids [7].

There is also evidence of their action in treating dermatological diseases such as psoriasis and other skin disorders, an example being that of Dead Sea peloids [10], in which it has been observed to reduce all skin symptoms of this disease (PASI index) [11] when combined with Dead Sea water and phototherapy. This water is furthermore observed to have antimicrobial action [12], an aspect of interest in the treatment of related dermatological alterations such as dermatitis. Other studies have shown that these muds can improve wound healing [13]. The effects of biogleas in thermal muds from Guardia Piemontese-Acquappesa were studied and found to significantly reduce desquamation, erythema, and itching in psoriasis [14].

Similar muds used in skin disorder applications are the Peruíbe muds [15] for psoriasis, dermatitis, acne, and seborrhoea. Peloids from Balaruc-les-Bains have been recently used for their anti-inflammatory, antioxidant, and healing properties [16]. Additionally, Spilioti et al. (2017) investigated the anti-inflammatory properties of 13 mud samples from Greek spa resorts by assessing their effect on the expression of the adhesion molecules ICAM-1 and VCAM-1 by endothelial cells as well as their effects on monocyte adhesion to activated endothelial cells. Most of the mud extracts used in the study inhibited TNF-a-induced expression of VCAM-1 by endothelial cells but showed little alteration on ICAM1 expression. Interestingly, the majority of the examined mud extracts markedly reduced monocyte adhesion to activated endothelial cells indicating a potent anti-inflammatory activity [17].

In terms of peloid composition for cosmetic and welfare purposes, many studies attribute their curative properties to their clay (less frequently peat or sapropels), mineral and trace elements content, and to the presence of microalgae and cyanobacteria.

### 2.1. Clays and Dermocosmetic Peloids

The use of clays in the preparation of peloids and dermocosmetic products have been studied by a great number of authors [1,3,18,19,20,21,22,23,24,25,26,27]. The main phyllosilicates present in most peloids are smectites, kaolinite, illite, illite–smectite mixed layers, and chlorite in different proportions [21].

Although there are fewer studies published on the composition of sapropels from Lake Techirghiol in Romania [28] and from lakes in Latvia, there are some studies that evaluate their potential medicinal and cosmetic use [29]. 

A comparative physico-chemical composition study of muds from different areas in the Homogeneous Euganean Hills Hydromineral Basin (B.I.O.C.E.) (Italy) reported the composition of peloids as “clayey-silt” (65.42% silt and 24.62% clay) and “silty-clay” (64.37% clay and 34.41% silt). Their heavy metal content was studied by comparison with commercial cosmetic mud and was found to be higher than in commercial mud; however, no allergic reactions were detected. A proposal to establish a protocol for effective control of these types of natural products has been put forward [30]. 

### 2.2. Minerals and Trace Elements in Dermocosmetic Peloids

Peloids for dermocosmetic and wellness applications are characterised by their varied composition in minerals and trace elements. The moisturising, soothing, and regenerating properties of the Dead Sea mud are attributed to a high magnesium content [31], which is well known for its anti-inflammatory and antiphlogistic effects and for its capacity to inhibit the polyamines involved in psoriasis pathogenesis [7]. Dead Sea mud also exhibits antimicrobial action, which is attributed to the high salt and sulphide concentrations plus its low pH, and it is therefore used in the treatment of acne [31].

In the case of the above-mentioned Peruíbe peloids, Da Silva-Cardoso et al. (2015) noted that the mud is enriched with Br, Cr, Sb, SE, and Zn ions during the maturation process and that these may be responsible for their anti-inflammatory properties [15,32]. 

### 2.3. Microalgae and Cyanobacteria in Dermocosmetic Peloids

One of the most outstanding and studied characteristics of peloids is their content in microalgae and cyanobacteria, which seem to exert a great influence on their cosmetic properties, since they have been proven to generate biologically active substances (especially during the maturation process), which in turn are responsible for the beneficial effects and actions on the skin [33]. 

There is abundant recent scientific literature on the biological fraction of peloids, and worth highlighting among them are studies on Euganea basin muds in the Spa area of Abano Terme (Italy). Thus, Ceschi-Berrini et al. in 2004 [34] described the presence of the genus *Phormidium* in thermal waters of the Euganea basin and subsequently identified the presence of acylglycerolipids produced by the aforementioned cyanobacteria, which appeared to confer therapeutic and cosmetic properties to the mud [35]. In a study of microbial diversity in the same area, Moro et al. (2007) [36] described a new species of Cyanoprokaryote called *Cyanobacterium aponinum* in the microbial mats of Euganean thermal springs. Subsequently, Poli et al. (2009) [37] described a thermophilic bacterium in the mud from this thermal basin that they called *Anoxybacillus thermarum*, which provides an idea of the special characteristics of the biological composition of these muds. Additional studies by Moro et al. (2010) expanded the biodiversity of these muds to species of the genus *Leptolyngbya* and *Spirulina* (now *Arthrospira*), suggesting that the cyanobacterial composition of phototrophic mats in the rather unusual environment of the Euganean Thermal District is variable, depending on the physico-chemical features of the different thermal spa waters. In fact, surveys carried out on 90 thermal spas suggest that the cyanobacterial diversity might be related to thermal mud processing in the different maturation tanks with thermal waters at different temperatures [38]. 

Research on the biological composition and organic matter present in the different maturation stages of Abano muds showed the presence of saturated and unsaturated fatty acids, hydroxyl acids, dicarboxylic acids, ketoacids, and alcohols and an increase in the lipid profile during the maturation process that peaked at six months. The presence of diatoms from clays was observed at the start of maturation; however, cyanobacteria belonging to the Oscillatoriales subsection progressively colonized the mud throughout maturation [39]. 

Centini et al. [40] recently analyzed the composition and antioxidant capacity of biogleas present in the Satunia Terme mud and confirmed earlier findings on the increased lipid profile during the maturation process and analyzed the hydrophilic fraction. Studies on antioxidant power revealed that bottom mud extracts are more active than surface extracts and that hydrophilic extracts are more active than lipid extracts. 

A comprehensive study using more than 650 mud samples from 29 places in the Abano area compared mineralogical and geological parameter variations with chlorophyll A in sludge during the mud maturation and recycling process. The conclusion was that chlorophyll A is converted into its derivatives and generates molecules that pass to the matured mud. Such a decrease in the chlorophyll A amount warrants maturation to take place in open tanks in order to maintain the photosynthetic process and to ensure that the amount of chlorophyll A and its derivatives continue to be sensitive to the supply of fresh mud [41]. Subsequent research by Gris et al. (2020) on the same muds (Euganean Thermal Muds) confirmed that the predominant species is *Phormidium* sp. and that diversity is greater when the temperature is 37–47 °C. At lower and higher temperatures, populations lose stability, thus exhibiting a significant change in species composition, low biodiversity, and low cyanobacterial abundance [42]. Zampieri et al. (2020) likewise noted the anti-inflammatory activity of exopolyssacharides from *Phormidium* sp present in the Abano muds [43]. 

Studies carried out on mud from Pausilya Therme di Donn’Anna (Italy) revealed antimicrobial capacity and identified seven taxa of green algae, two taxa of cyanobacteria, and even diatom taxa. In terms of the microalgae community, mud samples ripened for 6 months (6-month mud) presented a higher biodiversity compared to mud allowed to ripen for 1 month (1-month mud). The most abundant benthic microalgae taxa, identified in both samples and isolated exclusively from ripened mud, are *Chlorella* sp., *Coccomyxa* sp., *Scenedesmus* sp., *Leptolyngbya* sp., *Anabaena* sp., *Cocconeis placentula*, *Rhoicosphenia abbreviata* and *Navicula cincta*. *Nostoc* sp., *Scenedesmus* sp., *Chlamydomonas* sp., *Pseudococcomyxa simplex*, *Monodus* sp., *Gomphonema acuminatum*, *Amphora ovalis*, and *Nitzschia palea* [44].

In a like manner, the microbiological diversity of waters and muds from Sirmione Terme was characterised (using next-generation sequencing technology) by studying the different mud maturation stages: young (2-month old), intermediate (4-month old), and mature (6-month old). The results showed that three genera predominate: *Pelobacter*, *Desulfomonile*, and *Thiobacillus* and that *Pelobacter* levels increase during maturation while those of *Desulfomonile* and *Thiobacillus* decrease. The increase in phospholipid and sulpho- glycolipid fraction of mature muds reported by other authors [45] was attributed to *Pelobacter* by these authors. 

Muds from Balaruc-les-Bains (France) have also been analyzed to study the molecules responsible for their antioxidant, anti-inflammatory, and healing properties. Nine strains were analyzed and although no antioxidant activity was detected, a strong anti-inflammatory potential was observed for *Planktothricoides raciborskii*, *Nostoc* sp., and *Pseudo-chroococcus couteii*, and a slight wound-healing function was detected in extracts from *Aliinostoc* sp. [46], which is an activity of great interest in dermocosmetic and well-being treatments. Recent studies using morphological, ultrastructural, and molecular methods clearly identified the nine cyanobacterial isolates from the Thermes de Balaruc-Les-Bains muds as belonging to the orders Chroococcales: *Pseudochroococcus coutei*; Synechococcales: *Leptolyngbya boryana*; Oscillatoriales: *Planktothricoides raciborskii*, *Laspinema* sp., *Microcoleus vaginatus*, and *Lyngbya martensiana*; and Nostocales: *Nostoc* sp., *Aliinostoc* sp., and *Dulcicalothrix* sp. [47,48]. 

Dead Sea muds are well known for their use in the treatment of psoriasis [49]. They are high-salinity muds in which nine extremely halotolerant Bacillus species have been identified, one of them being *B. Paralicheniformis*, which confer a high antimicrobial action [50]. Subsequent studies have confirmed the antimicrobial property of *Bacillus persicusi* against different Gram+ and Gram− pathogens [51]. 

Organic fractions of mud from other environments have also been studied. Dolmaa et al. (2017) studied silty mud containing sulphide from Noggon Lake (Mongolia) and found that soluble organic matter contains a high percentage of hydrocarbons and their derivatives (33.68%) and that the lipid group contains fat-soluble vitamins including vitamins A, D, E, and their derivatives, plus steroids, which the authors relate to therapeutic and cosmetic properties [52].

Bigovic et al. (2019) examined the organic composition of Igalo Bay peloids (Montenegro), and they found them to mostly contain (saturated and unsaturated) fatty acids as well as essential amino acids, many of which have significant physiological, medical, and pharmaceutical properties [53].

Research carried out on the mineral peloids from Mariánské Lázne (Czech Republic) reported a new species of the genus *Aquitalea* (previously identified in humic lakes and peat marshes), which they called *Aquitalea pelogenes* (“derived or generated from mud”). They also found a profile of quinones and fatty acids upon analyzing the dry biomass. The polar lipids detected were diphosphatidylglycerol, phosphatidylethanolamine, phosphatidylglycerol, two unidentified phospholipids, and one unidentified aminophospholipid, to which the authors attributed the therapeutic properties [54].

Other studies reported changes in the microbial community composition of the peloid throughout maturation, wherein main changes take place in the early stages, with there being hardly any change between 3 and 6 months [55].

### 2.4. Safety of Peloids for Application in Dermocosmetics

Given that peloids are applied topically and in many cases on skin with dermatological disorders, their safety must be monitored for the possible presence of trace toxic metals and pathogenic microorganisms. 

Ma’or et al. (2015) studied the safety of Dead Sea muds used in cosmetics, by evaluating traces of nickel and chrome, and concluded that nickel and chrome concentrations measured in the mud are safe for human health insofar as systemic toxicity is concerned. They also observed that skin exposure to nickel and chrome is much lower since both metals mainly attach to the clay components in mud and are not easily released into the aquatic solution. The use of Dead Sea mud is not recommended for Ni^−^- or Cr^−^-sensitive persons [56].

Recently, Pavlovska et al. (2017) recommended testing in natural peloids (to be used as a raw material for pharmaceutical applications) not only heavy metals but also pesticides such as chlororganics, which are widely used as effective help to combat unwanted plant pests and pathogens and which have bioaccumulation and bioconcentration capabilities [29].

To ensure the quality and safety of the peloids, some properties should be determined; the most common are granulometry, plasticity, CEC and exchangeable cations, water content, pH, specific surface area, swelling power and swelling index, abrasiveness, density, rheological properties (viscosity), and thermal properties such as: specific heat capacity, thermal conductivity, thermal diffusivity, and thermal retentivity. For cosmetic uses could be also of interest to determine the parameters of hardness, springiness, adhesiveness, and cohesiveness, which are related to their visco-elastic properties [3]. From the microbiological quality and hygiene perspective, microbiological analyses such as total viable count (TVC), total coliforms, *E. coli*, enterococci, *S. aureus, P. aeruginosa*, and sulfite-reducing clostridia and dermatophytes fungi, must also be carried out [57].

## 3. Proposal for a Procedure to Manufacture Microalgae Peloids

This work puts forward a method that uses clays, microalgae/cyanobacteria, and mineral-medicinal water or seawater to manufacture peloids for use in cosmetics and in health and wellness programs at wellness centers.

Such peloids can be manufactured in the thermal spa itself for use with patients on the premises. Some examples of use in Europe are Abano Terme and Montecatini Terme (Italy), Dax Thermes, Eugenie-les-Bains or Barèges (France), Bad Bayersoien (Germany), and the thermal spas of Archena, Bohí, and El Raposo in Spain, these being mainly used to treat rheumatology and locomotor system disorders. Worth highlighting in Spain are the spas at Isla de la Toja and Balneario de Compostela that manufacture their own peloids for use in dermatology and dermocosmetics, with interesting results in psoriasis and dermatitis [58,59]. In such cases, the peloid is considered as a spa product derived from mineral-medicinal water and is governed by the spa legislation of each country. An example is shown in Figure 6.

However, if the product is destined for marketing as a cosmetic product, it is governed by cosmetic regulations. REGULATION (EC) No 1223/2009 OF THE EUROPEAN PARLIAMENT AND OF THE COUNCIL of 30 November 2009 on cosmetic products (https://eur-lex.europa.eu/legal-content/EN/TXT/?uri=CELEX:02009R1223-20190813 accessed on 21 October 2021) defines the stages involved in the manufacture and marketing of a cosmetic product in Europe, including the lifecycle of a cosmetic product, from its conception in R&D laboratories to the monitoring of its effects and effectiveness after marketing.

### 3.1. Composition of a Peloid

A peloid is comprised of a solid fraction or substrate made of clays, sediments, or peat and a liquid fraction made of mineral-medicinal water, seawater, or brackish/salt-lake water, and it may contain a biological fraction from the water or the solid substrate [2,8]. When manufacturing peloids for dermocosmetic purposes, one should use high-quality clays to guarantee safety and effectiveness on skins, which in many cases are damaged. Their composition is shown in Figure 7.

#### 3.1.1. Solid Substrate: Clays

As indicated earlier, the solid component of a peloid can be diverse. In order to achieve good thermo-physical characteristics and applicability, we propose the use of clays containing smectite (bentonite) and kaolinite, since the former have very good plastic properties [21] and the latter help regulate skin secretions and the final pH of the mixtures [8].

#### 3.1.2. Solid Substrate: Mineral-Medicinal Water and Seawater

Each mineral-medicinal water is unique and thus the first step is gaining knowledge of its chemical composition, including the majority and trace elements, as well as physico-chemical characteristics such as pH, electrical conductivity, and the possible presence of dissolved gases. 

All mineral-medicinal waters must be analyzed periodically to guarantee quality before and during their application in spas, as provided for in the legislation of the different countries. This is why all of them are analyzed and quality is guaranteed. However, given that often only the major elements are analyzed, it is of utmost importance to analyze the trace elements when developing peloids for cosmetic us since their dermocosmetic potential lies in them [60]. Table 1 summarizes the principal majority and trace elements in mineral-medicinal waters of interest for the manufacture of cosmetic products.

#### 3.1.3. Microalgae and Cyanobacteria

They are one of the differential components in a peloid; and given that each mineral-medicinal water is unique, one needs to study the type of microalgae/cyanobacteria present therein. The plankton composition in seawater differs through latitudes unlike the composition of seawater, which is similar at all latitudes, and hence one needs to study the type of microalgae present in a particular environment.

We therefore propose that microalgae/cyanobacteria cultures be sourced from the mineral-medicinal waters or seawater, by means of a process adapted to the characteristics of each species or genus predominant therein. The culture process includes growth in an appropriate medium (mineral-medicinal or seawater) with the necessary nutrients and light stimulation depending on the type of species (Figure 8).

### 3.2. Preparation of a Dermocosmetic Peloid with Microalgae

The process of preparing a peloid with microalgae or cyanobacteria involves a few prior stages in which raw materials are first studied before carrying out tests on the mixtures. The stages are summarised in Figure 9 and in the following subsections: (i) selection of raw materials (clays, thermal waters, and microalgae cyanobacteria); (ii) characterization of raw materials (different test and determinations to asess its suitability and optimal properties); (iii) preparation and testing mixtures (using different proportions of the raw materials); (iv) characterization of the peloid sample (including maturation process if necessary); and (v) use and effectiveness test.

#### 3.2.1. Selection of Raw Materials

Raw materials or initial materials (clays, microalgae, and waters) are selected for the intended purposes. Given that the peloid is intended for dermocosmetic and/or welfare uses, the clays selected must be of a high quality and have an affinity for the skin (kaolinites, bentonites, etc.). The water used is the one present at the spa: mineral-medicinal water (or seawater), and the microalgae/cyanobacteria can either be those present at the thermal spa or the thalassotherapy center, but others acquired lyophilized or frozen can also be used.

#### 3.2.2. Characterization of Raw Materials

All raw materials must be properly characterised. The most frequent tests performed on clays are mineralogical analysis; chemical composition; granulometry; SEM study; swelling; cation exchange capacity and exchangeable cations; percentage of water, solids, and ash; and differential thermal analysis and thermogravimetry [6,21,61,62,63].

The spa water or seawater must also be analyzed to study the majority and trace elements, in addition to other physico-chemical analyses. The most important parameters are temperature, electrical conductivity, dry residue, turbidity, cations and anions, dissolved gases, radioactivity, hardness, and alkalinity. One also needs to study properties such as density, thermal conductivity, specific heat, viscosity, and thermal diffusivity [63,64,65]. 

It is furthermore important to characterize microalgae or cyanobacteria and undertake studies to isolate and obtain a mono-specific and clonal culture. The sample is characterized through a chemical analysis, determination of crystalline phases, and by studying its composition (proteins, lipids, carbohydrates, vitamins, etc.) [66]. 

#### 3.2.3. Preparation and Testing of Mixtures

Mixtures are prepared using different proportions of the three raw materials and tested for texture, spreadability, ease of application, etc.

The mixtures are then selected, characterized, and subject to use and efficacy tests. 

#### 3.2.4. Characterization of the Peloid Sample

The most common analyses carried out on the sample of the selected peloid or mixture are density, thermal conductivity, specific heat, viscosity, rheological behavior, and thermal diffusivity [62,65,67,68]. For a peloid to be suitable for pelotherapy uses, it should have several properties, such as a low cooling rate, a high absorption capacity, a high cationic exchange capacity, good adhesiveness, handling easiness, and a pleasant feeling when applied to the skin. Among all the above properties, the cooling rate is one of the most critical ones, since the heat contributed by the peloid also plays a role as a therapeutic agent. In many therapeutic applications, therefore, the peloid must be kept at a higher temperature than that of the patient’s body during application [6]. 

If peloids need maturing, then one must also establish the temperature, light, agitation, etc. conditions. In any case, the characterization analyses are the same, and samples need to be taken after 15 days, 1 month, 2 months, etc. until the maturation process is complete and no further changes in the physico-chemical parameters are observed [39,55,61,69,70]. 

#### 3.2.5. Use and Effectiveness Tests

Different analyses and tests are carried out on volunteers to evaluate user acceptance of the peloid and its effectiveness. Inclusion and exclusion criteria for both tests must be established, taking into account that these preliminary studies are carried out on healthy persons. Additional controlled clinical trials must be done if the peloid is finally destined to treat skin conditions such as psoriasis, dermatitis, etc.

The use test consists of a set of questions related to texture, ease of application, sensations during and after application, skin condition after product removal, etc. In Figure 10, an example of a microalgal peloid is shown.

Efficacy studies are usually objective determinations done through skin biometrology techniques, such as hydration (by corneometry), grade of sebum (with sebumeter), skin elasticity (cutometry or elastometry), and, sometimes, transepidermal water loss [71,72,73].

## 4. Conclusions

Peloids have been used for therapeutic purposes since time immemorial, mainly in the treatment of locomotor-system pathologies and dermatology. Their effects are attributed to their components, i.e., to the properties and action of mineral waters, clays, and their biological fraction, which may be made up of microalgae, cyanobacteria, and other organisms present in water and clays. Different studies show that the biological fraction and the maturation process (in which components remain in contact for a certain length of time) contribute to the formation of biologically active compounds.

Even though there are many studies on the therapeutic use of peloids made with microalgae/cyanobacteria, very little research has been done on dermocosmetic applications. Such research demonstrates their potential as soothing, regenerating, antioxidant, anti-inflammatory, and antimicrobial agents. Their effect is related to the presence of unsaturated fatty acids, acylglycerolipids, sulfoglucolipids, vitamins, alcohols, phenols, etc., as well as sulphur derivatives, minerals (Ca, Mg, etc.), and trace elements (Zn, Se, Si, etc.).

Each thermal spa has a unique natural mineral water with specific physico-chemical characteristics, which are the basis of their therapeutic actions (along with other mechanisms related to the application technique). Moreover, specific microbiota consisting mainly of microalgae and/or cyanobacteria are often found in it. This is why thermal spas, thalassotherapy centres, and wellness centres in general should progress towards making their own dermocosmetic products using their natural mineral water or seawater; a solid substrate, preferably clay; and the microalgae/cyanobacteria. Hence, a method for the manufacture of a dermocosmetic peloid was presented based on the experience of the authors and existing publications, with indications for its characterization and efficacy study. 

## Figures and Tables

**Figure 1 marinedrugs-19-00666-f001:**
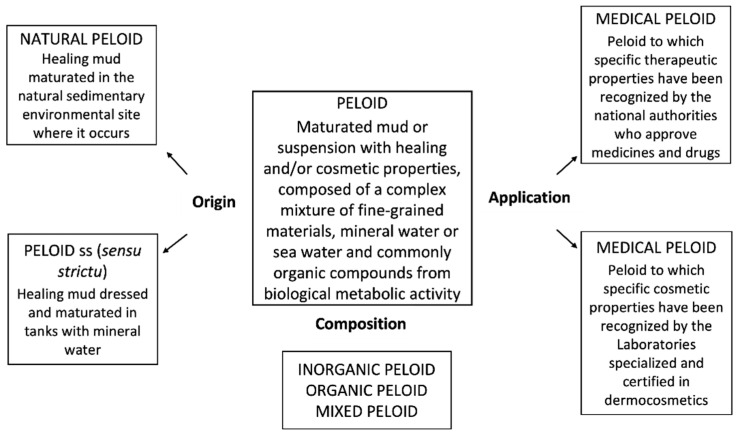
Peloid classification with regard to origin, composition, and applications (from Gomes et al. 2013).

**Figure 2 marinedrugs-19-00666-f002:**
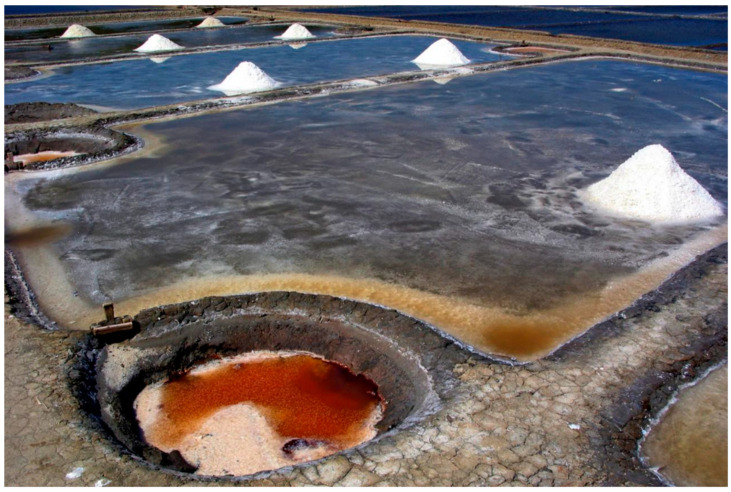
Natural maturation peloid (saline mud from Sečovlje salt pans, Sečovlje Nature Park, Slovenia).

**Figure 3 marinedrugs-19-00666-f003:**
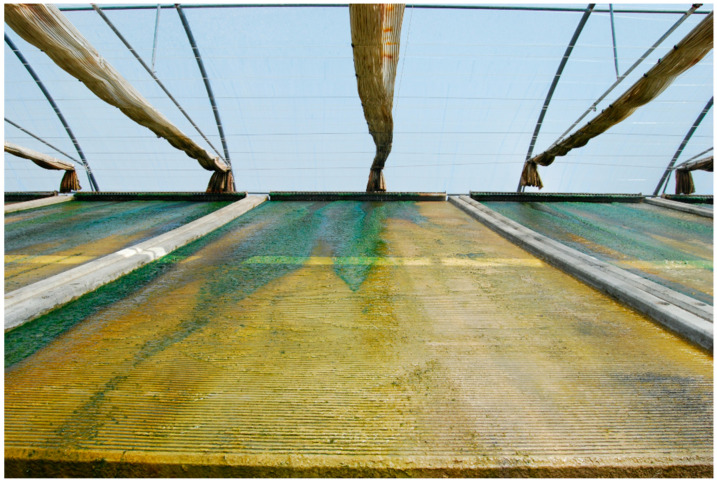
Cyanobacteria cultivation for peloid preparation (Dax, France).

**Figure 4 marinedrugs-19-00666-f004:**
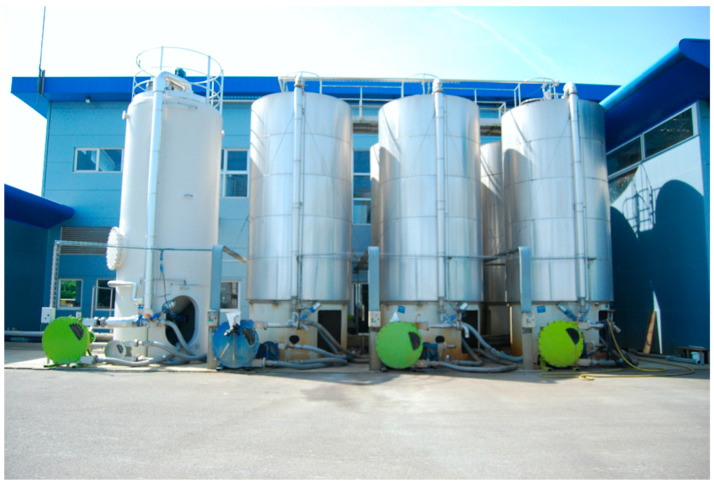
Tanks for artificial peloid maturation (Dax, France).

**Figure 5 marinedrugs-19-00666-f005:**
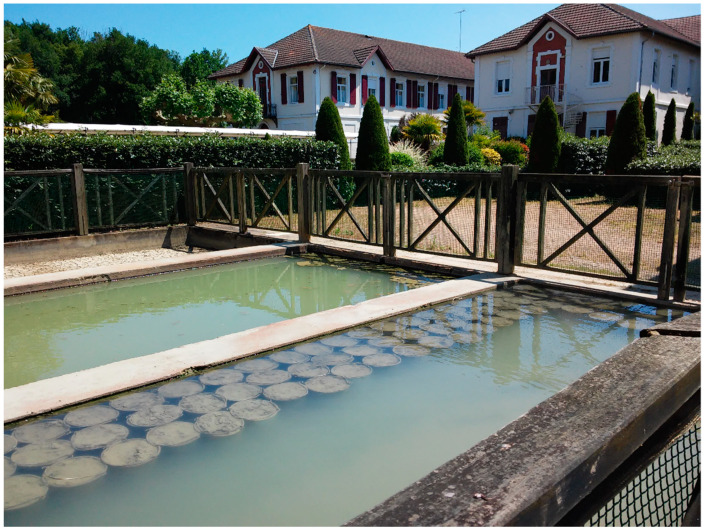
Thermal mud maturation (Dax, France).

**Figure 6 marinedrugs-19-00666-f006:**
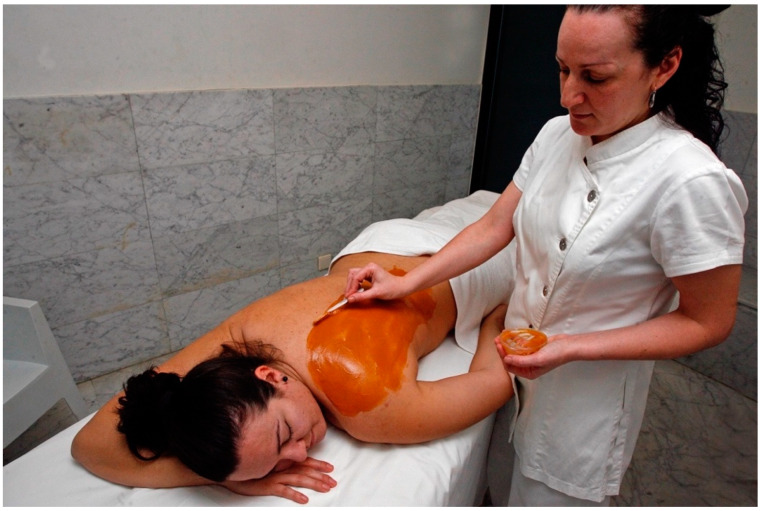
Manufactured peloid; application for psoriasis and dermatological conditions (La Toja thermal spa, Pontevedra, Spain).

**Figure 7 marinedrugs-19-00666-f007:**
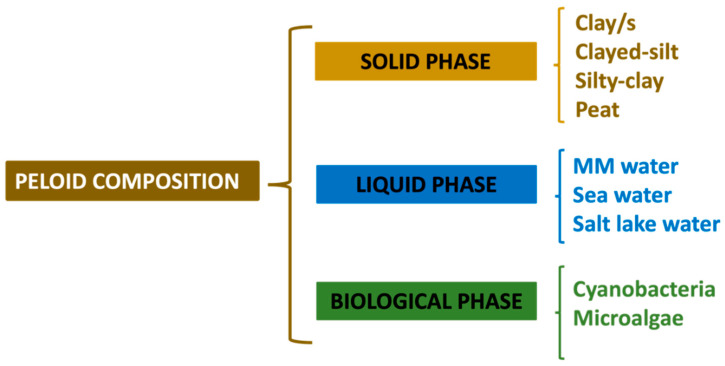
Composition of a peloid (MM: mineral-medicinal).

**Figure 8 marinedrugs-19-00666-f008:**
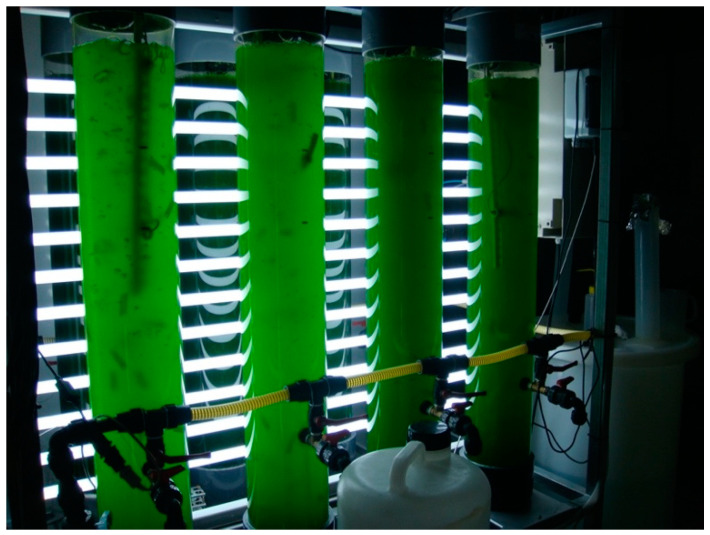
Photobioreactor with microalgae (cultivation at FA2 lab; Applied Physics Department; University of Vigo).

**Figure 9 marinedrugs-19-00666-f009:**
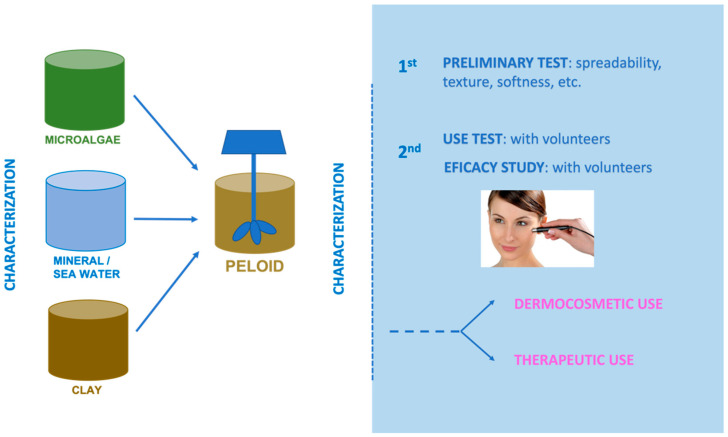
Peloid manufacture: procedure and test.

**Figure 10 marinedrugs-19-00666-f010:**
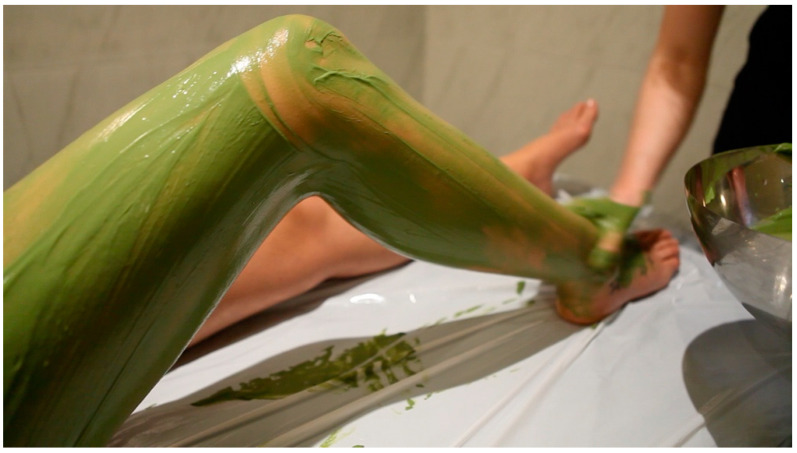
Application of microalgal peloid (Talaso Atlántico, Baiona, Pontevedra, Spain).

**Table 1 marinedrugs-19-00666-t001:** Majority and trace elements in mineral-medicinal waters that have an effect on the skin (Mourelle & Gómez, 2015).

Chemical Element	Effect on the Skin
Calcium	Effect on proteins that regulate cell divisions: calmodulin and cellular retinoic-acid-binding protein (CRAB) Catalysing action of differentiation enzymes: transglutaminase, protease, and phospholipases Indispensable for regulating permeability of cell membranes Regulation of proliferation and differentiation of keratinocytes
Sulphur	Cell regenerator, keratolytic/keratoplastic (dose-dependent) Antibacterial, antifungal
Magnesium	Inhibits synthesis of some polyamines involved in psoriasis pathogenesis at concentrations of 5 × 10^−4^, and its reduction by magnesium improves disease condition Anti-inflammatory, antiphlogistic Catalyses synthesis of nucleic acids and proteins Catalyses ATP production Produces sedation in the central nervous system
Chloride	Fluid balance of tissues
Sodium	Fluid balance of tissues
Copper	Anti-inflammatory, immune system maintenance
Chromium	Enzymatic activator
Fluorine	Energy supply in keratinocytes
Manganese	Immune system modulator
Nickel	Stimulates cell development in tissues
Zinc	Antioxidant; prevents ageing; healing and regeneration of skin tissues
Silicon	Involved in collagen and elastin synthesis and cell metabolism Present in colloidal silica form in many mineral waters used in dermatology Has a dermoabrasive and emollient effect on psoriatic plaques

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
