# Peer review of "Microalgal Peloids for Cosmetic and Wellness Uses"

_marinedrugs, 2021, doi:10.3390/md19120666_

Round 1
Reviewer 1 Report
Marine Drugs (marinedrugs-1451784)
Title: Microalgal peloids for cosmetic and wellness uses
Comments to the authors:
The submitted review discussed the peloids and their therapeutic purposes. They were used since time immemorial, mainly in the treatment of locomotor system pathologies and dermatology. Their effects were attributed to their components, i.e. to the properties and action of mineral waters, clays and their biological fraction, which may be made up of microalgae, cyanobacteria and other organisms present in water and clays. The authors proposed a method for the manufacture of a dermocosmetic peloid.
I think the review can be accepted for publication after the authors respond to the following comments:
- The review is disorganized. The authors have to reorganize the whole review into clear sections with clear messages, The authors have to establish a coherent story for their review.
- Some paragraphs are made one or two sentences which is uncommon in scientific manuscripts. The authors have to restructure their sentences to write coherent paragraphs made of several sentences carrying a single idea.
- The authors have to indicate the search engines used to collect their information.
- The authors should indicate the keywords used to search for relevant literature.
- The authors should indicate the time span covered by their review.
- The authors should indicate their contribution to the field that allowed them to write a review on peloids.
- The proposed method for preparation is not clear and cannot be easily followed. The authors should rephrase the method.
- All figures are irrelevant. Why did the authors put a figure with the life cycle of cosmetic products? The figure with microalgae does not give any useful information. Did the authors take this photo from their laboratory? If they own the photo they have to give more information.
- The authors should add photos of peloids to give the reader a better understanding of the material.
Author Response
The review is disorganized. The authors have to reorganize the whole review into clear sections with clear messages, The authors have to establish a coherent story for their review.
The paper is divided into 4 sections and several subsections. We kindly ask for a further explanation.
- Some paragraphs are made one or two sentences which is uncommon in scientific manuscripts. The authors have to restructure their sentences to write coherent paragraphs made of several sentences carrying a single idea.
Some sentences have been rewritten.
- The authors have to indicate the search engines used to collect their information.
SciFinder, PubMed, Web of Science, Scopus.
- The authors should indicate the keywords used to search for relevant literature.
Keywords are: “pelotherapy”, “mud therapy”, “peloids and skin; “thermal mud”, “microalgae and thermal water,” “cyanobacteria and thermal water”, “mud and cosmetics”, “mud and dermocosmetics”; “mineral water and skin”; “seawater and skin”.
- The authors should indicate the time span covered by their review.
Up to October 2021.
- The authors should indicate their contribution to the field that allowed them to write a review on peloids.
Publications in the field of characterization of peloids; and microalgae. Some examples are:
- Gomes, C.; Carretero, M.I.; Pozo, M.; Maraver, F.; Cantista, P.; Armijo, F.; Legido, J.L.; Teixeira, F.; Rautureau, M.; Delgado, R. Peloids and pelotherapy: Historical evolution, classification and glossary. Clay Sci. 2013, 75, 28–38.
- Legido, J.; Medina, C.; Mourelle, M. L.; Carretero, M.; Pozo, M. Comparative study of the cooling rates of bentonite, sepiolite and common clays for their use in pelotherapy. Clay Sci. 2007, 36, 148–160.
- Carretero, M. I.; Pozo, M.; Legido, J.L.; Fernández-González, M. V.; Delgado, R.; Gómez, I.; Armijo, F.; Maraver, F. Assessment of three Spanish clays for their use in pelotherapy. Clay Sci. 2017, 99, 131–143.
- Glavaš, N.; Mourelle, M. L.; Gómez, C. P.; Legido, J. L.; Šmuc, N. R.; Dolenec, M.; Kovac N. The mineralogical, geochemical, and thermophysical characterization of healing saline mud for use in pelotherapy. Clay Sci. 2017, 135, 119–128.
- Legido, J.L.; Medina, C.; Mourelle, M.L.; Carretero, M.I.; Pozo, M. Comparative study of the cooling rates of bentonite, sepiolite and common clays for their use in pelotherapy. Clay Sci. 2007, 36, 148–60.
- Casas, L.M.; Pozo, M.; Gómez, C.P.; Pozo, E.; Bessieres, D.; Plantier, F.; Legido, J.L. Thermal behavior of mixtures of bentonitic clay and saline solutions. Appl Clay Sci. 2013, 72, 18–25.
- Casas, L.M.; Legido, J.L.; Pozo, M.; Mourelle, L.; Plantier, F.; Bessieres, D. Specific heat of mixtures of bentonitic clay with sea water or distilled water for their use in thermotherapy. Acta. 2011, 524, 68–73.
- Mato, M. M.; Casas, L.M.; Legido, J.L.; Gómez, C.P.; Mourelle, L.; Bessieres, D.; Plantier, F. Specific heat of mixtures of kaolin with sea water or distilled water for their use in thermotherapy. Therm. Anal. Calorim. 2017, 130, 479–484.
- Mourelle, M. L., Carmen P. Gómez, and José L. Legido 2017. "The Potential Use of Marine Microalgae and Cyanobacteria in Cosmetics and Thalassotherapy" Cosmetics4, no. 4: 46. https://doi.org/10.3390/cosmetics4040046
Several project to develop peloids from natural mineral water and sea water: 4 projects with Thermal Spa El Raposo (Spain); 1 project La Toja Thermal Spa (Spain); 1 project Compostela Thermal Spa (Spain); 1 project Guitiriz Thermal Spa (Spain); 1 project Talaso Atlántico (Spain);
Organization of Iberoamerican Congress on Peloids (7 editions).
President / secretary of the following societies: Galician Society of Peloids (SOGAPETER) and Spanish Society of Peloids (SEPETER); Iberoamerican Society of Peloids (SIPET).
- The proposed method for preparation is not clear and cannot be easily followed. The authors should rephrase the method.
Method improved.
- All figures are irrelevant. Why did the authors put a figure with the life cycle of cosmetic products? The figure with microalgae does not give any useful information. Did the authors take this photo from their laboratory? If they own the photo they have to give more information.
Information added.
- The authors should add photos of peloids to give the reader a better understanding of the material.
Photos added.
Thank you very much for your suggestions and advice.
Reviewer 2 Report
The review presented by authors deal with the cosmetic use of peloids as "nutra-cosmeceutical" ingredients. I suggest not to use such a redundant term in the title; it has no cosmetic significance even if its use is widespread.
Even if many cases of application have been reported, a classification of different available natural peloids according their mineral/chemical composition, or according their characterizing parameters is missing.
The aim of the review it is not clear, because at first authors deeply discuss the application of natural peloids and then they propose an "artificial" one.
An aspect never considered is the discussion of the potential risks for human health. Peloids may contain elements with toxicological relevance that could be released during application.Taking into account the importance of safety nowadays, I think that a paragraph regarding this aspect would be an essential key, in order to drive towards the use of self made peloids and to reinforce the motivation of the last part of the work.
Which peloids are the most common used in cosmetics? (a general discussion about the market is needed... sulphurous? bromine monochloride-iodine peloids?)
I think that a paragraph related to the most common analysis to ensure quality and safety has to be inserted to complete the work.
The review focuses only on the cosmetic and therapeutic activity of peloids. I think this is simplistic. The review offers a few points of reflection and results not helpful for the cosmetic insiders. It needs to be extended
Author Response
The review presented by authors deal with the cosmetic use of peloids as "nutra-cosmeceutical" ingredients. I suggest not to use such a redundant term in the title; it has no cosmetic significance even if its use is widespread.
Nutra-cosmeceutical is not mentioned in the title. We kindly ask for a further explanation.
Even if many cases of application have been reported, a classification of different available natural peloids according their mineral/chemical composition, or according their characterizing parameters is missing.
Classification added.
The aim of the review it is not clear, because at first authors deeply discuss the application of natural peloids and then they propose an "artificial" one.
Explanation added.
An aspect never considered is the discussion of the potential risks for human health. Peloids may contain elements with toxicological relevance that could be released during application.Taking into account the importance of safety nowadays, I think that a paragraph regarding this aspect would be an essential key, in order to drive towards the use of self made peloids and to reinforce the motivation of the last part of the work.
Subsection 2.4 improved.
Which peloids are the most common used in cosmetics? (a general discussion about the market is needed... sulphurous? bromine monochloride-iodine peloids?)
Information added.
I think that a paragraph related to the most common analysis to ensure quality and safety has to be inserted to complete the work.
Information added.
The review focuses only on the cosmetic and therapeutic activity of peloids. I think this is simplistic. The review offers a few points of reflection and results not helpful for the cosmetic insiders. It needs to be extended
Information added.
Thank you very much for your suggestions and advice.
Reviewer 3 Report
The review which is written by Mourelle et al. is rather intresting for the dermocosmetics research. According to my opinion this work deserves a publication in Marine Drugs. However, there are some minor changes that need to be done before.
Line 21, 25. Please replace the word “centres” with “centers”.
Line 30. Replace precisely with precise
Line 38. it appears you have an extra space between “slit peloids”. The same between the words “is from”
Line 50. Please remove own
Line53. Replace “in an empirical manner” with the word “empirically”
Line 55. Please replace the word “ageing” with “aging”
Line 61 . Replace the word biogleas with “ biogas”
line 71. Authors miss another interesting work related to use of clays. Please add the work of Spilioti et al. 2017, Environmental geochemistry and health. There are not enough works in this area and it worths to be mentioned as many as it can be.Line 123. Replace “colonised” to “colonized”
Line 124/126/290. Replace analysed to analyzed
Line 154. Replace “characterised” to characterized
Line 180. Add “ a” before high percentage
Line 192. Replace analysing to analyzing
Line 283/345 Replace centre to center
Line 284. Relace lyophilised to lyophilized
Line 306. Replace behaviour to behavior
Line 309/348. Replace characterisation to characterization
Please check reviews for a format consistency according to Marine drugs recommendations.
Author Response
All the suggestions added (except replacement “biogleas”).
Thank you very much for your suggestions and advice.
Round 2
Reviewer 2 Report
Authors have implemented the manuscript according to my comment
I think the work can be accepted as it is